# Extracellular Vesicles Are Important Mediators That Regulate Tumor Lymph Node Metastasis via the Immune System

**DOI:** 10.3390/ijms24021362

**Published:** 2023-01-10

**Authors:** Yoshitaka Kiya, Yusuke Yoshioka, Yuichi Nagakawa, Takahiro Ochiya

**Affiliations:** 1Department of Molecular and Cellular Medicine, Tokyo Medical University, 6-7-1 Nishi-Shinjuku, Shinjuku-ku, Tokyo 160-0023, Japan; 2Department of Gastrointestinal and Pediatric Surgery, Tokyo Medical University, 6-7-1 Nishi-Shinjuku, Shinjuku-ku, Tokyo 160-0023, Japan

**Keywords:** extracellular vesicles, tumor metastasis, lymph node metastasis, premetastatic niche, microenvironment, immune system, immune tolerance

## Abstract

Extracellular vesicles (EVs) are particles with a lipid bilayer structure, and they are secreted by various cells in the body. EVs interact with and modulate the biological functions of recipient cells by transporting their cargoes, such as nucleic acids and proteins. EVs influence various biological phenomena, including disease progression. They also participate in tumor progression by stimulating a variety of signaling pathways and regulating immune system activation. EVs induce immune tolerance by suppressing CD8^+^ T-cell activation or polarizing macrophages toward the M2 phenotype, which results in tumor cell proliferation, migration, invasion, and metastasis. Moreover, immune checkpoint molecules are also expressed on the surface of EVs that are secreted by tumors that express these molecules, allowing tumor cells to not only evade immune cell attack but also acquire resistance to immune checkpoint inhibitors. During tumor metastasis, EVs contribute to microenvironmental changes in distant organs before metastatic lesions appear; thus, EVs establish a premetastatic niche. In particular, lymph nodes are adjacent organs that are connected to tumor lesions via lymph vessels, so that tumor cells metastasize to draining lymph nodes at first, such as sentinel lymph nodes. When EVs influence the microenvironment of lymph nodes, which are secondary lymphoid tissues, the immune response against tumor cells is weakened; subsequently, tumor cells spread throughout the body. In this review, we will discuss the association between EVs and tumor progression via the immune system as well as the clinical application of EVs as biomarkers and therapeutic agents.

## 1. Introduction

Extracellular vesicles (EVs) are secreted by various types of cells and have a lipid bilayer structure. They are found in blood, urine, breast milk, semen, and other bodily fluids. EVs contain mRNAs, noncoding RNAs, DNA fragments, proteins, and lipids. EVs transport these cargoes from donor cells to recipient cells; consequently, they are involved in intercellular communication and the biological transformation of recipient cells [1].

In 1946, Chargaff et al. reported that molecules similar to thromboplastic protein were purified from human blood by high-speed centrifugation, and these molecules recovered coagulation dysfunction. These purified factors were predecessors to EVs [2]. In 1967, Wolf et al. purified microparticles from plasma by ultracentrifugation, and these vesicles were rich in phospholipids and had coagulant properties resembling those of Platelet Factor 3. The authors called these vesicles “platelet dust” and defined them as subcellular coagulant materials [3]. In the 1970s–1980s, accumulating evidence indicated the presence of microparticles in plasma, cell culture supernatants, and cancerous ascites [4,5]. In 1983, Pan and Johnstone et al. observed that sheep reticulocytes secreted vesicles that were approximately 100 nm in size and had a lipid bilayer structure, but they considered these vesicles to be only a system for eliminating unnecessary intracellular material [6]. In 1987, these microparticles were named “exosomes” [7]. In 1996, Raposo et al. revealed that B cell-derived exosomes induced an antigen-specific MHC class II-restricted T-cell response, and they showed that the exosomes that had been identified as a system for eliminating cellular waste were actually associated with immune activation [8]. Moreover, subsequent studies have revealed that dendritic cell-derived exosomes express MHC class I and II on their surface and suppress the growth of established murine tumors by stimulating T-cell activation [9,10]. Since the 2000s, clinical trials have been carried out to determine whether dendritic cell-derived exosomes are efficient as cancer vaccines [11,12,13]. A phase II clinical trial was investigated the clinical benefit of exosomes that were derived from IFN-γ-matured dendritic cells and loaded with MHC class I- and II-restricted cancer antigens as maintenance immunotherapy after induction chemotherapy in patients with inoperable non-small cell lung cancer without tumor progression. The primary endpoint, which was at least 50% progression-free survival at 4 months after chemotherapy cessation, was not reached. However, this phase II trial confirmed the ability of dendritic cell-derived exosomes to enhance the NK cell arm of antitumor immunity in patients with advanced NSCLC [13].

Subsequently, it was elucidated that exosomes not only express biological molecules such as membrane proteins on their surface but also contain various proteins and nucleic acids, and they transport their cargoes and influence the biological functions of recipient cells [14,15,16]. Moreover, several researchers have given various names to these vesicles according to their size and secretory mode [17]. The International Society for Extracellular Vesicles (ISEV), established in 2011, recommended referring to these microparticles as extracellular vesicles because a standardized name for these vesicles was needed. In 2014, the ISEV presented “minimal experimental requirements for definition of extracellular vesicles and their functions”. A list of minimal details for studies of extracellular vesicles (MISEV 2014) outlined procedures for EV separation/isolation, characterization, and functional studies [18]. Moreover, in 2018, the ISEV presented MISEV 2018, which provided clearer explanations of the importance of each recommendation and highlighted the extent of author consensus on each section [19]. Following this suggestion, we collectively call these microparticles “EVs” in this article.

EVs mediate biological interactions between various cells by transporting their cargoes. Additionally, EVs also influence the development or progression of diseases, including cancer, by stimulating various signaling pathways and regulating immune system activation. Accumulating evidence has shown that EVs play an important role in promoting tumor malignancy (proliferation, migration, and invasion), altering the microenvironments of primary lesions or premetastatic sites, destroying the defense mechanisms against tumor dissemination (peritoneum, blood–brain barrier, and so on), inducing drug resistance, and suppressing immune responses [20,21,22,23,24].

Here, we review the relationship between EVs and malignant disease from an immunological perspective because EVs are closely associated with immune system regulation. For example, we will describe that EVs mediate tumor metastasis by interacting with tumor-adjacent tissues or premetastatic sites, especially lymph nodes, which are organs that regulate immune responses. In addition, EVs, which are secreted by all types of cells throughout the body, including tumor cells, reflect the characteristics of the donor cells and express specific molecules, such as tumor-specific antigens. Thus, we will explain that EVs are expected to act as clinical biomarkers or therapeutic devices.

## 2. EVs Mediate Cancer Development

### 2.1. EVs Enhance the Malignancy of Tumor Cells

Malignant tumor progression occurs due to the induction of proliferation, migration, and invasion. These processes are controlled by a complex network of various signal transduction pathways, such as the Wnt signaling pathway, PI3K/Akt/mTOR pathway, Ras/Raf/MEK pathway, receptor tyrosine kinase pathway, and transforming growth factor (TGF) receptor pathway. These signaling pathways are stimulated by signaling molecules that bind to cell surface receptors or enter the cytoplasm by crossing the plasma membrane. Additionally, EVs secreted from tumor cells or cells in the tumor microenvironment stimulate signaling pathways in tumor cells themselves or in adjacent cells. Previous studies have shown that secreted EVs usually promote tumor malignancy by communicating with recipient cells in an autocrine or paracrine manner [25] (Figure 1). Tumor cell-derived EVs participate in tumor cell proliferation in an autocrine/paracrine manner; for example, chronic myeloid leukemia (CML)-derived EVs contain large amounts of TGF-β1, which activates the ERK, Akt, and NF-κB pathways by binding to the TGF-β1 receptor on the surface of CML cells. The activation of these pathways regulates the proliferation and survival of tumor cells [26]. Epiregulin, which is an epidermal growth factor receptor (EGFR) ligand, is enriched in metastatic salivary adenoid cystic carcinoma-derived EVs [27]. These EVs stimulate the EGFR on tumor cells in an autocrine/paracrine manner and enhance the motility and invasiveness of tumor cells [28]. Similarly, tumor cell-derived EVs influence the transformation or characteristics of tumor cells [29]. It is widely known that EVs induce epithelial–mesenchymal transition (EMT) by upregulating N-cadherin expression or downregulating E-cadherin and GLI-1 expression; thus, EVs enhance the motility and invasiveness of tumor cells [30,31]. For example, hypoxia-inducible factor-1α in EVs promotes the tumor metastasis of nasopharyngeal carcinoma by regulating the expression of E-cadherin and N-cadherin, which are associated with EMT [32]. EVs that are secreted by oral squamous cell carcinoma (OSCC) cells under hypoxic conditions contain miR-21, which increases the migration and invasion of normoxic OSCC cells in a HIF-1α- and HIF-2α-dependent manner; miR-21 significantly decreases the E-cadherin levels in OSCC cells, thus promoting EMT [33,34].

Tumor cell-derived EVs also modify the biological phenotypes of cells that are adjacent to local lesions in an autocrine/paracrine manner; that is, EVs activate receptors or regulate gene expression in recipient cells. Nedawi et al. described that only a small percentage of glioma cells express EGFR variant III (EGFRvIII). EVs containing this receptor are transported to EGFRvIII-negative cells and fuse with their plasma membranes, thereby sharing this receptor. Consequently, EV-mediated cargo transport promotes oncogenic activity, that is, activation of transforming signaling pathways (MAPK and Akt), changes in the expression of EGFRvIII-regulated genes (VEGF, Bcl-xL, p27), morphological transformation, and increased anchorage-independent growth capacity [35]. In addition, several studies have shown that EVs from highly malignant tumor cells transfer their specific mutant genes or molecules, such as mutant KRAS in colon cancer, long noncoding RNA (lncRNA) ZFAS1 in gastric cancer, or esophageal squamous cell carcinoma (ESCC), and lncRNA ZEB1-AS1 in ESCC, to wild-type cells [36,37,38].

Tumor-derived EVs also promote tumor motility by reprogramming mesenchymal cells in the tumor microenvironment. miR-9, which is upregulated in various breast cancer cell lines and was identified as a prometastatic miRNA, affects the properties of human breast fibroblasts, promoting the switch to the cancer-associated fibroblast (CAF) phenotype, via tumor-derived EVs. Moreover, miR-9 is also secreted by CAFs and alters tumor cell behavior by modulating its direct target E-cadherin and fibroblast cells themselves [39]. Some studies of CAF-derived EVs have been reported. miR-181d-5p contained in CAF-derived EVs in breast cancer promotes proliferation, invasion, migration, and EMT and inhibits apoptosis of cancer cells by targeting caudal-related homeobox 2 (CDX2) and downregulating CDX2 and their downstream gene homeoboxA5 (HOXA5) [40]. miR-500a-5p in CAF-derived EVs in breast cancer promotes proliferation and metastasis ability by targeting and reducing ubiquitin-specific peptidase 28 (USP28) [41].

A number of studies have shown that EVs enhance cancer malignancy. Importantly, regarding the association of EVs with cancer progression, EVs can comprehensively influence the tumor microenvironment, which is composed of various cell populations, rather than only mediating communication between cells of the same specific type because, although they are secreted from a particular tumor, EVs are heterogeneous vesicle populations. Although we discussed that EVs enhance tumor malignancy by interacting directly with tumor cells or tumor microenvironmental cells in this section, EVs can also promote tumor progression indirectly by influencing distant organs or the immune system. Thus, EVs can mediate not only local effects around tumors in an autocrine/paracrine manner but also systemic effects via blood circulation in an endocrine manner. We have reviewed how EVs promote tumor metastasis in distant organs next section.

### 2.2. EVs Promote the Formation of a Premetastatic Niche and Facilitate Tumor Metastasis

Several investigations have elucidated that tumor cells or adjacent mesenchymal cells affect the microenvironment in distant organs via EVs and that EVs promote the development and survival of tumor cells in these distant organs. Thus, EVs induce various sequential events in premetastatic organs that collectively result in the establishment of a premetastatic niche. These events include the enhancement of vascular permeability and angiogenesis, conversion of fibroblast cells to CAFs, recruitment of immune cells, such as bone marrow-derived cells (BMDCs), and suppression of immune responses; consequently, the premetastatic niche promotes the prometastatic potential of tumor cells [42,43,44] (Figure 2). Moreover, EVs exhibit different patterns of integrin expression that are associated with corresponding organotropic characteristics; for example, α6β4 and α6β1 are associated with lung metastasis, and αvβ5 is associated with liver metastasis [45]. After arriving at the target organ, EVs mobilize BMDCs, such as macrophages, neutrophils, and mast cells, and promote the establishment of a premetastatic niche [46]. EVs from highly metastatic melanomas increase the metastatic behavior of primary tumors by permanently educating bone marrow progenitors through the receptor tyrosine kinase MET. Metastatic melanoma-derived EVs also enhance vascular permeability at premetastatic sites and reprogram bone marrow progenitors to acquire a provasculogenic phenotype [47]. EVs derived from pancreatic ductal adenocarcinoma (PDAC) are taken up by Kupffer cells and cause TGF-β secretion and increased fibronectin production in hepatic stellate cells. The fibrotic change causes the recruitment of bone marrow-derived macrophages into the microenvironment. PDAC-derived EVs upregulate the expression of macrophage migration inhibitory factor (MIF), and MIF knockdown inhibits liver premetastatic niche formation and metastasis [48]. Similarly, tumor cell-derived EVs also suppress the immune response by recruiting BMDCs or tumor-associated macrophages to the premetastatic niche [24]. Tumor cell-derived EVs express heat shock protein 72 (Hsp72) on their surface, and the interaction between Hsp72 and myeloid-derived suppressor cells (MDSCs) determines the suppressive activity of MDSCs via the activation of STAT3 [49,50]. There are two different main populations of macrophages: classical tumor suppressive macrophages (M1) and alternative tumor promotive macrophages (M2). Tumor-derived EVs induce the functional polarization of macrophages toward the M2 phenotype and thus promote an immunosuppressive environment in the premetastatic niche [51,52].

On the other hand, Plebanek et al. described that poorly metastatic melanoma cell-derived EVs potently inhibit metastasis to the lung. These nonmetastatic EVs stimulate an innate immune response through the expansion of Ly6C^low^ patrolling monocytes (PMo) in the bone marrow, which cause cancer cell clearance at the premetastatic niche via NK cell recruitment and the TRAIL-dependent killing of melanoma cells by macrophages [24].

It is certain that tumor-derived EVs influence the microenvironment in premetastatic organs, but there are many different and complicated mechanisms. Several studies indicated the possibility that EVs could function as potential diagnostic or prognostic biomarkers but also suggested that accumulating further evidence is needed to establish their value as biomarkers. Elucidating the components of EVs or the specific factors by which EVs interact with recipient cells might provide information about the organotropic nature of metastasis or the immunosuppressive nature of tumors in individual patients, which may lead to the development of more appropriate therapies.

## 3. EVs Mediate the Immune Response to Cancer

### 3.1. EVs Suppress the Anticancer Response by Immunoediting

The immune response that is associated with the development or progression of tumor cells has been well studied. The biological immune system commonly eliminates tumors that arise de novo and inhibits the progression of tumor cells via immune surveillance.

Previous studies have shown that the immune system prevents tumor development in three ways. First, the immune system can protect the host from virus-induced tumors by eliminating or suppressing viral infections. Second, the immune system can prevent the establishment of inflammatory conditions related to tumorigenesis by eliminating pathogens or resolving inflammation. Third, the immune system can identify and eliminate tumor cells in certain tissues on the basis of their expression of tumor-specific antigens. This third mechanism is known as cancer immune surveillance, and this process identifies transformed cells that have evaded tumor-suppressor mechanisms and eliminates them before they can establish tumor lesions [53]. Although the immune system acts as a tumor suppressor, tumor cells can spread in the body after a certain point. Schreiber et al. proposed a concept named cancer immunoediting, and this concept describes the relationship between the immune system and tumor development. The authors described successive alterations in tumor cells based on this idea [54].

Cancer immunoediting includes three distinct phases: elimination, equilibrium, and escape. The elimination phase is immune surveillance; in this phase, immune cells identify the developing tumors and eliminate them before their clinical appearance. Although tumor cells exhibit high immunogenicity in the early period of tumor development, their immunogenicity is gradually decreased by immunoediting. The immune system sometimes cannot completely eliminate these variant tumor cells; thus, surviving variant tumor cells enter the equilibrium phase, where the adaptive immune system inhibits tumor progression. In this equilibrium phase, tumor cells can lie dormant and might remain clinically undetected for the life of the host. Finally, tumor cells enter the escape phase due to further characteristic changes that occur in the tumor cells by immunoediting or due to the collapse of the immune system with aging. These events lead to the end of tumor dormancy and the clinical appearance of tumor lesions [53,55].

Tumor lesions progress according to such a process, and tumor cell-derived EVs are also associated with the immune system in various ways, for example, by suppressing the proliferation of CD8^+^ T cells [56,57] or by expressing the ligands that bind to death receptors, such as Fas and TNFα, and thus inducing T-cell apoptosis [58]. These studies revealed that tumor cells can escape from the immune system by suppressing immune cell attack or inducing immune cell apoptosis via EVs.

On the other hand, several studies have demonstrated that tumor-derived EVs also activate immune cells by inducing the secretion of inflammatory cytokines or promoting tumor progression or metastasis. miR-21 and miR-29a in lung cancer-derived EVs bind to TLR8 and activate immune cells, such as macrophages, inducing the activation of NF-κB or the secretion of prometastatic inflammatory cytokines [59]. Breast cancer-derived EVs stimulate the activation of NF-κB in macrophages and promote the secretion of inflammatory cytokines, such as IL-6, TNFα, GCSD, and CCL2, thus inducing the progression of diseases [60].

Furthermore, tumor-derived EVs induce the transformation of immune cells from a tumor-eliminating phenotype to tumor-promoting phenotype; thus, EVs derived from ovarian cancer, ESCC, glioblastoma, and other cancers induce the polarization of macrophages toward the M2 phenotype via various pathways. [51,61,62,63].

Pucci et al. showed that melanoma-derived EVs disseminate via lymphatics and preferentially bind to subcapsular sinus (SCS) CD169^+^ macrophages in tumor-draining lymph nodes. The CD169^+^ macrophage layer physically blocks tumor-derived EV dissemination, but the barrier is weakened by tumor progression or the side effects of therapeutic agents. Although the authors did not evaluate the mechanism for weakening the macrophage barrier, they showed that a disrupted SCS macrophage barrier enables tumor-derived EVs to enter the LN cortex, interact with B cells, and facilitate tumor-promoting humoral immunity [64] (Figure 3).

According to the immunoediting theory, tumor lesions can clinically appear when the equilibrium between tumor progression and immune response is disrupted. Unfortunately, therapeutic agents, such as chemotherapies, might induce this disequilibrium. Even though cytotoxic chemotherapeutic agents damage tumor cells, they might also decrease immune activity. This situation might induce distant tumor metastasis before the complete elimination of tumor cells. Moreover, much attention has been drawn to immunotherapy, especially immune checkpoint inhibitors, which can restore the host immune system and induce the apoptosis of tumor cells via CD8^+^ T cells.

Thus, we will review the association between immune checkpoint inhibitors, which have grown in importance in recent years, and EVs in the next section.

### 3.2. EVs Participate in the Expression of Immune Checkpoint Molecules

Immune checkpoint molecules are receptors that inhibit the excessive activation of T cells and prevent autoimmunity, and representative receptors include PD-1, CTLA-4, and TIM3. Under normal physiological conditions, activated T cells release IFN-γ and upregulate PD-L1 in adjacent cells. Upregulated PD-L1 decreases T-cell activity and inhibits damage to self-tissues. It is now clear that during tumor progression, tumor cells can coopt immune checkpoint molecules to evade immune recognition [65,66].

Immune checkpoint molecules are involved in tumor development or progression. Several studies have revealed that these molecules promote tumor progression by inducing immune evasion or, in contrast, inhibit tumor development by suppressing chronic inflammation [67,68]. In addition, it was reported that EVs derived from activated T cells restore immune surveillance by inhibiting immune checkpoint molecules on the surface of tumor cells [69].

When IFN-γ upregulates PD-L1 expression on the surface of tumor cells, PD-L1 is also expressed on the surface of tumor-derived EVs and suppresses T-cell activation and proliferation [70,71]; furthermore, there is a correlation between PD-L1 expression levels in tumor tissues and those in EVs [72,73]. EVs regulate the expression of PD-L1 in tumor lesions in vitro and vivo, and administration of EVs that carry PD-L1 can restore the proliferation of PD-L1-knockout tumor cells [72,74,75,76]. Several studies have shown that tumor-derived EVs expressing PD-L1 are associated with tumor stage or prognosis [77,78]. This evidence supports the clinical application of EVs, and EVs are expected to act as biomarkers. We will review the clinical application of EVs later.

## 4. EVs Are Associated with the Development of the Lymphatic System and Tumor Lymphatic Metastasis

### 4.1. EVs Support Lymphangiogenesis

Lymphatic vessels begin to form during fetal development via the development of lymphatic endothelial cells (LECs) from the anterior cardinal vein [79]. In the early stages of fetal development, endothelial cells in the anterior cardinal vein differentiate into LECs, and LECs start to express biological molecules such as LYVE-1 and Prox1 [80,81,82]. Subsequently, LECs express vascular endothelial growth factor receptor 3 (VEGFR-3) and podoplanin, which is a ligand of CLEC2, a platelet activation receptor. VEGF-C secreted from stromal cells induces LECs that express high VEGFR-3 levels to sprout from the venous wall, which results in the establishment of primary lymph sacs [83]. Then, the cardinal vein and the primary lymph sacs are divided, and peripheral lymphatic vessel formation is induced by the continuous sprouting of LECs. At the incomplete division phase, blood flow can reach the primary lymph sacs, and CLEC-2 binds to podoplanin on LECs, which induces platelet activation and the release of platelet granule contents. TGF-β family proteins in the granules suppress the migration and proliferation of LECs and promote the division of blood vessels and lymphatic vessels [84,85,86]. The development of lymphatic vessels is associated with various biological molecules, and the representative molecules mentioned above are used as specific markers of lymphatic vessels in experimental fields [80,82].

The generation of new lymphatic vessels from existing lymphatic vessels is called lymphangiogenesis, and this process sometimes occurs secondary to diseases; thus, this sequential process of sprouting, progression, and hyperplasia of lymphatic vessels is induced by pathological conditions, such as inflammation, tissue repair, and tumor dissemination [87]. Previous studies have shown that several growth factors, such as members of the VEGF family, platelet growth factor-BB (PDGF-BB), fibroblast growth factor 2 (FGF2), and hepatocyte growth factor (HGF), are involved in lymphangiogenesis; however, the effect of these molecules, except VEGF family members, is often mediated by indirect processes that induce VEGF-C/D [88,89,90].

There have been various studies on the relationship between lymphangiogenesis and malignant diseases, including the mechanism underlying tumor-induced lymphangiogenesis with VEGF-C/D and LYVE-1 [91,92], the mechanism underlying tumor-induced lymphangiogenesis before tumor metastasis in sentinel lymph nodes [93,94], and the correlation between increased tumor-induced lymphangiogenesis and poor prognosis [95]. These studies have indicated that tumor lymph node metastasis follows organized processes. First, lymphangiogenesis is induced in sentinel lymph nodes near the primary lesion, which enhances the transfer of tumor cells to draining lymph nodes, and finally, lymph node metastasis is promoted. Various studies have indicated the relationship between tumor-derived EVs and lymphangiogenesis, which is the first step of lymph node metastasis.

miR-221-3p is found in EVs derived from cervical squamous cell carcinoma (CSCC), and it is associated with the expression of LYVE-1. Zhou et al. showed that CSCC-derived EVs promote human LEC (HLEC) migration and tube formation in vitro and facilitate lymphangiogenesis and lymph node metastasis in vivo. These authors also showed that vasohibin (VASH) is a negative regulator of lymphangiogenesis and a target of miR-221-3p; furthermore, the miR-221-3p-VASH axis activates the ERK/Akt pathway in HLECs [96]. Ultraconserved RNA 189 (uc.189) in ESCC-derived EVs targets EPHA2 in LECs and promotes lymphangiogenesis by activating the P38MAPK/VEGF-C pathway [97]. Similarly, several cargoes of EVs, such as the heparin-binding factor midkine in melanoma-derived EVs [98], ELNAT1 in bladder cancer-derived EVs [99], and the lncRNA HANR in hepatocellular carcinoma-derived EVs [100], promote lymphangiogenesis by inducing VEGF-C or VEGFR-3. These cargoes directly or indirectly influence the expression level of VEGF-C, which is the key player in lymphangiogenesis, but there were also some EV cargoes that do not affect the expression of VEGF-C. miR-320b is significantly upregulated in the ESCC-derived EVs of patients who developed lymph node metastasis. miR-320b promotes lymphangiogenesis by activating Akt signaling in HLECs but does not affect the expression of VEGF-C [101]. In addition, the lncRNA LNMAT-2, which is upregulated in EVs derived from lymph node metastasis-positive bladder cancer, is taken up by HLECs and enhances lymphangiogenesis or lymph node metastasis in a VEGF-C-independent manner by upregulating PROX expression [102].

These studies indicated that tumor-derived EVs promote lymphangiogenesis and lymph flow from the primary lesion to draining lymph nodes, which facilitates the development of lymph node metastasis.

### 4.2. EVs Support the Premetastatic Niche at Lymph Nodes

During the establishment of lymph node metastasis, the transformation of the microenvironment in the metastatic lymph node occurs prior to tumor cell proliferation at that site [43,103]. This transformed microenvironment is referred to as the premetastatic niche. Premetastatic niches are characterized by stromal changes such as differentiation of myofibroblast cells, remodeling of extracellular matrix, angiogenesis or activation of endothelial cells, and recruitment of BMDCs [104,105]. Previous studies have indicated that VEGFR-1-positive myeloid-derived precursor cells form colonies before tumor cells arrive at the lymph nodes, and lymphangiogenesis is induced at this site [94,106,107]. Then, tumor cells encounter immune cells in the lymph nodes, and these cells interact with each other via EVs and soluble factors; thus, the immune response to tumor cells is mediated by these interactions [108,109]. During the recruitment of immune suppressive cells to the lymph nodes, immune cells, such as myeloid-derived suppressor cells, tumor-associated macrophages, regulatory T cells, and immature dendritic cells, are mobilized and suppress the activation of CD4^+^ or CD8^+^ T cells and NK cells; as a result, these immune suppressive cells promote tumor metastasis [110,111]. The S100A8 and S100A9 proteins, which are derived from melanoma, inhibit the maturation of dendritic cells, which enables the primary tumor to establish a premetastatic niche in the draining lymph nodes. These immature dendritic cells are present in the sentinel lymph nodes before lymph node metastasis becomes clinically apparent [112,113]. Therefore, the immune response in the lymph node establishing a premetastatic niche might be suppressed, which thus induces lymph node metastasis. Analysis of the regional lymph nodes of esophageal cancer revealed a decisive difference in gene expression patterns in the nonmetastatic regional lymph nodes between lymph node metastasis-positive patients and lymph node metastasis-negative patients. These regulated genes included DKK1, which is a Wnt pathway inhibitor that is associated with enhanced inflammatory responses, antigen presentation, reduced cellular growth, immune cell tracking, and cell-to-cell signaling; thus, the immune response is suppressed in lymph node metastasis-positive patients [114]. Here, we have described esophageal cancer, but premetastatic niches are also established in many other cancers. Nogues et al. described how tumor-derived EVs are associated with the establishment of a premetastatic niche in lymph nodes in a review article [115]. Christopher et al. suggested that EVs derived from bladder cancer might influence premetastatic niche formation in lymph nodes. These authors showed that Tenascin-C is upregulated in the benign lymph nodes of lymph node metastasis-positive bladder cancer patients and that bladder cancer-derived EVs might induce premetastatic niche formation in lymph nodes by directly targeting tenascin-C [105]. Although these reports about premetastatic niche formation in lymph nodes are highly significant, they are limited by a small sample size, and the observations require validation in a larger series. Moreover, it would be important to determine the prognostic significance of tenascin-C expression in lymph nodes strictly within nonmetastatic patients.

### 4.3. EVs Influence Lymphatic Endothelial Cells and Contribute to Cancer Immune Evasion

In Section 4.1, we described the association between EVs and lymphangiogenesis, and in Section 4.2, we described the role of tumor-derived EVs in the establishment of a premetastatic niche in lymph nodes. In this section, we summarize how EVs induce tumor immune tolerance by influencing LECs or immune cells in lymph nodes. This occurs because lymphatic systems are involved in transporting bodily fluid, antigens, and immune cells from peripheral tissues back into circulation via lymph nodes, where immune surveillance occurs and adaptive immune responses are initiated. In addition, LECs importantly participate in establishing immune tolerance [116,117,118,119,120]. LECs suppress the maturation of dendritic cells and subsequent adhesion-dependent CD8^+^ T-cell priming [116,120]. In addition, LECs modulate peripheral T-cell tolerance by presenting endogenously expressed tissue-specific antigens via MHC class I molecules and eliminating autoreactive CD8^+^ T cells [117,118,119,121]. LECs also induce the apoptosis of antigen-specific CD8^+^ T cells by cross-presenting exogenous antigens via MHC class I molecules [122]. Altogether, LECs perform an antigen-presenting function and induce immune tolerance to endogenous or exogenous antigens [122]. In particular, lymph node-resident LECs (LN LECs) regulate T-cell activity via immune checkpoint molecules on the surface of T cells [117,123]. LN LECs constitutively express PD-L1, which suppresses T-cell activity by binding to the PD-1 receptor. PD-L1 expression by LN LECs is regulated by lymphotoxin signaling in the lymph node microenvironment [124]. During tumor development, tumor-associated LECs induce immunoregulation and facilitate tumor metastasis [125]. Several studies have shown that these tumor-associated LECs inhibit the antitumor T-cell response in the tumor microenvironment. In a study of an orthotopic murine model of melanoma, tumor-associated LECs induced dysfunction of CD8^+^ T cells by cross-presenting a tumor-derived antigen and thus promoted tumor expansion [126]. Dieterich et al. showed that PD-L1 expression is upregulated in LECs, which results in T-cell inhibition in tumor mouse models [127]. The expression of PD-L1 in LECs is upregulated by the secretion of IFN-γ from tumor stromal cells, tissue-infiltrating antigen-specific CD8^+^ T cells, and so on; consequently, LECs inhibit T-cell accumulation in tumors [127,128]. EVs were also found to induce immune tolerance by influencing LECs. Zhou et al. showed that miR-142-5p in CSCC-derived EVs is taken up by LECs, and it causes CD8^+^ T-cell exhaustion by upregulating the expression of indoleamine 2,3-dioxgenase (IDO) [129]. These authors also showed that miR-1468-5p in CSCC-derived EVs upregulates PD-L1 expression in human dermal LECs (HDLECs), thus inducing immunosuppression by promoting the exhaustion of CD8^+^ T cells. miR-1468-5p induces the reprogramming of HDLECs by suppressing the expression of HMBOX-SOCS1 and activating the JAK2/STAT3 pathway, allowing cancer cells to escape antitumor immunity [130]. Leary et al. found that melanoma-derived EVs selectively interact with lymph node-resident macrophages and LN LECs, which induces the remodeling of lymph nodes and alters the transcriptional profile of LN LECs; that is, EVs induce the establishment of a premetastatic niche in lymph nodes. Furthermore, EVs transfer tumor-specific antigens to draining lymph nodes, and LN LECs cross-present antigens and induce the apoptosis of tumor-specific CD8^+^ T cells [131] (Figure 4).

Lymphatic systems are closely related to tumor progression. Lymphatic vessels enhance the transfer of tumor cells by lymphangiogenesis, and LECs also induce immune tolerance to tumors by suppressing T-cell activity. Although several studies have evaluated whether tumor-associated LECs inhibit T-cell activity via immune checkpoint molecules, the detailed mechanism underlying the induction of immune tolerance to tumors by LECs is not well understood. In addition, EVs have been investigated due to their function in the blood circulation, but their interaction with lymphatic systems has rarely been reported. Further elucidation of the induction of immune tolerance to tumors by LECs and the interaction between EVs and lymphatic systems might accelerate the progress of immunotherapy.

## 5. Clinical Application of EVs

### 5.1. EVs Might Be Potential Diagnostic Biomarkers

Tissue biopsies are usually performed for diagnosis, staging, and monitoring therapeutic effects; in contrast, liquid biopsies have grown in importance in recent years. Liquid biopsy is minimally invasive, samples are easily obtained, and this approach is faster and more economical than tissue biopsy. EVs, cell-free DNAs (cfDNAs), and circulating tumor cells (CTCs) are present in liquid biopsies. EVs secreted from tumor cells reflect the intracellular status of the donor cells, and tumor-derived EVs are secreted in large amounts from early lesions. Thus, real-time detection of the changes in EV cargoes could provide important information for accurate diagnosis, prognosis, and disease monitoring [132]. EVs have a lipid bilayer structure, and the stability of the biological molecules in EVs is high; hence, EVs and their cargoes could be stably preserved for over 90 days under general storage conditions [133]. cfDNAs and CTCs that are associated with cancer development and progression are uncertain and limited compared with tumor-derived EVs. Additionally, they are rarely present in blood circulation until tumor sizes increase to some degree; therefore, it is difficult to use these molecules for early diagnosis [132,134,135]. In fact, various studies indicate an association between EVs and diagnostic potential or prognosis prediction, so nucleic acids in EVs or membrane proteins on the surface of EVs might be novel biomarkers [136,137,138,139,140,141,142,143,144,145,146,147,148,149,150,151,152,153]. In this review, we have summarized that EVs mediate the immune evasion of tumors and induce tumor metastasis; below, we will show that EVs might be a diagnostic biomarker of tumor metastasis or an indicator of the immune response to tumors. As described in the previous section, LECs that are influenced by tumor-derived EVs suppress T-cell activity via immune checkpoint molecules and induce lymph node metastasis. Thus, EVs are a strong link between the immune system and lymph node metastasis (Table 1). Some studies have indicated the efficacy of the use of EVs as diagnostic markers of lymph node metastasis. For example, ESCC is well known to have a high potential for lymph node metastasis; hsa_circ_0026611 and uc.189 are expressed in ESCC-derived EVs, and these molecules correlate with lymph node metastasis of ESCC [97,154]. The serum level of EV-derived hsa_circ_0026611 in patients with ESCC with lymph node metastasis is significantly higher than that in patients with ESCC without lymph node metastasis. Thus, serum EV-derived hsa_circ_0026611 levels could be used as a significant parameter to discriminate patients with lymph node metastatic ESCC from patients without lymph node metastasis with an area under the curve (AUC) of 0.724 [154]. uc.189 in EVs promotes the proliferation, migration, and tube formation of HLECs by activating the P38MAPK/VEGF-C pathway by binding to EPHA2 [97]. Thus, EV-associated molecules can be used as clinical application tools for the diagnosis of lymph node metastasis. The ability to diagnose lymph node metastasis might facilitate prognostic prediction because lymph node metastasis is a poor prognostic factor in various malignant diseases. Other studies suggested that EVs could help to identify appropriate treatments. In glioblastoma with EGFRvIII mutation, EGFRvIII is also observed in circulating EVs. EVs can help to confirm EGFR mutation status and make decisions about therapy because EGFRvIII-positive glioblastomas are over 50 times more likely to respond to EGFR inhibitor treatment [155]. Some studies have indicated that EVs might predict sensitivity to treatments. For example, androgen receptor splice variant 7 (AR-V7) was correlated with resistance to hormonal therapy in patients with castration-resistant prostate cancer. AR-V7 is also expressed in EVs derived from these patients, so these EVs might become a predictive biomarker of therapeutic resistance [156]. Another study showed that PD-L1 expressed in tumor-derived EVs might be a predictor of sensitivity to immune therapy. The expression levels of PD-L1 on the surface of melanoma-derived EVs might reflect the status of antitumor immunity. A high level of PD-L1 in EVs before treatment might reflect the extreme exhaustion of T cells; consequently, T cells might not be reactivated by PD-1 treatment. In contrast, for on-treatment patients, the increase in PD-L1 levels in EVs due to T-cell activation could reflect a good therapeutic effect [71].

These studies have indicated the possibility that EVs could be used as biomarkers for therapeutic decision making, prediction of therapeutic effects, and various forms of clinical support; however, these molecules have not been used as clinical tools because of the complication of isolating EVs. In most studies, EVs were isolated by the ultracentrifugation of serum or other body fluids, and the protocols varied substantially; hence, an easier and more effective isolation and detection approach is needed.

### 5.2. EVs Might Be Potential Therapeutic Devices

EVs are expected to be novel therapeutic agents because of their association with tumor development. There are some therapeutic strategies that inhibit tumor progression by targeting EVs, such as the elimination of circulating tumor-derived EVs, the suppression of EV secretion from tumor cells, or the disruption of EV absorption by recipient cells [157]. One study examined a method of eliminating tumor-derived EVs with hemofiltration, which can effectively remove tumor-promoting EVs by targeting specific antibodies to the surface of EVs [158]. Nishida-Aoki et al. proposed a novel therapeutic strategy for eliminating tumor-derived EVs tagged with antibodies. In this article, EVs that were incubated with anti-CD9 or anti-CD63 antibodies were preferentially internalized and eliminated by macrophages in vitro and in vivo [159].

Recently, substantial attention has been given to the use of immune checkpoint inhibitors as a novel therapy for various cancers [160,161,162,163,164,165,166,167,168]. In PD-L1-expressing tumor patients, PD-L1 is also expressed on the surface of secreted EVs and induces resistance to immune checkpoint therapy by binding to anti-PD-L1 antibodies. It was suggested that decreasing the numbers of these EVs might restore the therapeutic effects of immune checkpoint inhibitors [169,170]. Poggio et al. showed that eliminating EVs by knocking out Rab27 or nSMase2, which are factors related to EV biogenesis, inhibits tumor progression in vivo because tumor cells secrete a vast majority of PD-L1 molecules in EVs rather than maintaining PD-L1 molecules on their cell surface. Moreover, these PD-L1-deficient tumor cells suppressed the growth of wild-type tumor cells that express PD-L1 at distant sites. Thus, these results indicated that suppressing local EV secretion could induce a systemic immune response against multiple tumor sites simultaneously [169]. Wang et al. focused on ferroptosis, which contributes to antitumor immune effects, and they developed a nanounit constructed with GW4869, an inhibitor of EV biogenesis, and Fe^3+^, an inducer of ferroptosis. The nanounit stimulated T-cell activity and enhanced the response to anti-PD-L1, so these authors suggested that the combination of the nanounit and a PD-L1 inhibitor could be a next-generation cancer immunotherapy [170].

Previous studies have described phase I and phase II clinical trials of immunotherapy with EVs [11,12,13]. In a phase II trial, peripheral blood mononuclear cells (PBMCs) extracted from patients were differentiated into dendritic cells, and dendritic cell-derived EVs (DEX) were administered as a cancer vaccine to patients. As a result of the phase II trial, administering the cancer vaccine to chemotherapy-stabilized or responding patients increased NK cell function; however, it did not induce T-cell activation. The primary endpoint, which was at least 50% progression-free survival at 4 months after chemotherapy cessation, was not reached because highly immunosuppressive tumors inhibited the antitumor effects of DEX and upregulated PD-L1 expression on the surface of DEX [13].

Furthermore, several studies have shown that EVs can act as a vehicle for drug delivery, and studies have evaluated the cytotoxicity of EVs loaded with antitumor drugs [171,172,173,174]. The efficacy of chemotherapy drugs was increased via EVs because integrins and glycans on the surface of EVs influenced the organotropism of EVs, and autologous tumor-derived EVs allowed immune evasion and the absorption of drugs by target cells [172,175].

In addition, a study with EV-mimetic nanovesicles derived from M1 macrophages showed that the nanovesicles suppressed tumor progression by repolarizing M2 macrophages to M1 macrophages in the tumor microenvironment [176].

Although these studies suggested the possibility that EVs could be used for clinical applications, further accumulating evidence is needed regarding the pharmacological safety and efficacy of EVs in the human body. Some studies have revealed a relationship between EVs and immune checkpoint molecules, and this topic attracted substantial attention in recent years; thus, EVs might be novel key players in immunotherapy. Further evidence could provide novel therapeutic options and improve the prognosis of lethal cancers.

## 6. Conclusions

In this review, we addressed the effect of EVs on tumor progression from various perspectives. Previous studies revealed that EVs are mediators of intercellular communication and induce changes in tumor cells and the tumor microenvironment by transporting their cargoes and EVs also promote tumor metastasis by distant organs or the immune system. Recently, tumor immunity has attracted substantial attention, and it was indicated that EVs are closely associated with tumor immunity, inducing immune tolerance to tumors and the expression of immune checkpoint molecules. It was suggested that EVs act as a bridge between tumor immunity and tumor progression, including lymph node metastasis, so they might have clinical applications. Thus, we expect that EVs can be novel therapeutic agents including immunological effects for cancer, which are completely different from existing therapeutic agents. We also expect EVs to have a preventive effect on tumor metastasis of highly metastatic tumors because of their close association with establishing a premetastatic niche. Although a number of studies describing the efficacy of EVs have been reported, these applications of EVs have not yet been introduced into clinical practice because there are some problems, for example, how to isolate EVs and analyze target molecules as well as confirmation of the safety and efficacy of EV therapy. Moreover, EVs are heterogeneous vesicle populations that are secreted by various cells in the body, and they promote tumor progression as a result of the confluence of various different factors; thus, the use of specific EVs alone might not be effective. Further analysis of EV-associated molecules and a study of EVs with a large sample size are needed to support the clinical use of EVs.

## Figures and Tables

**Figure 1 ijms-24-01362-f001:**
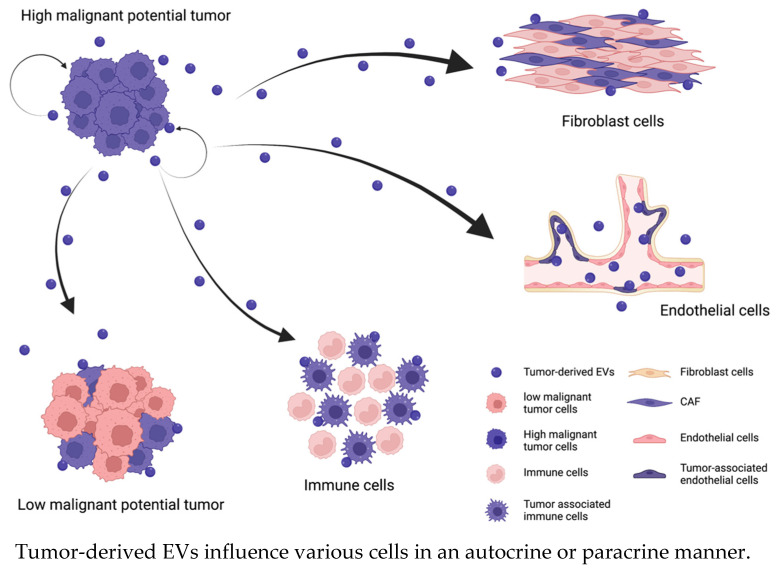
Tumor-derived EVs enhance the malignancy of the cells in same tumor and alter the characteristics of cells in the tumor microenvironment in an autocrine/paracrine manner. EVs regulate various signaling pathways by interacting with recipient cells or binding to receptors on their surface. The recipient cells regulated by tumor-derived EVs enhance their malignancy (changing the cell color pink to purple) and promote tumor progression.

**Figure 2 ijms-24-01362-f002:**
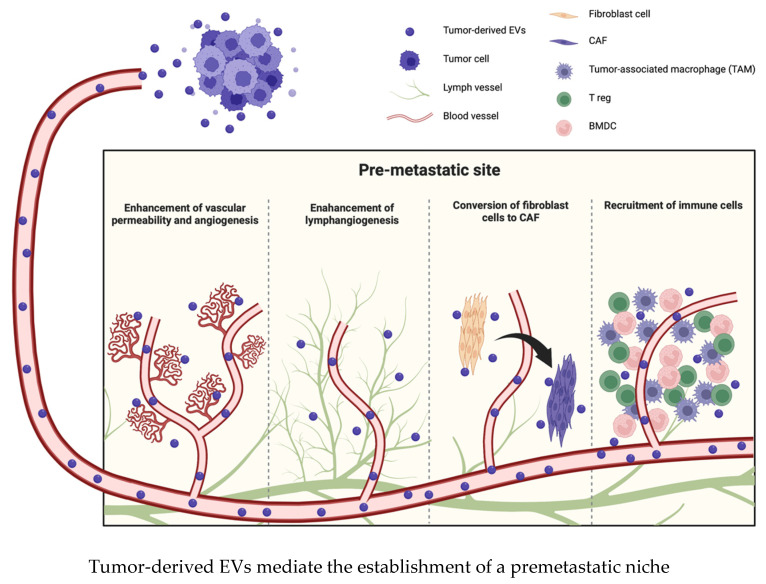
EVs transform the microenvironment in the premetastatic site before metastatic lesions appear. EVs induce various changes, such as the enhancement of vascular permeability, angiogenesis or lymphangiogenesis, conversion of fibroblast cells to CAFs, and recruitment of immune cells.

**Figure 3 ijms-24-01362-f003:**
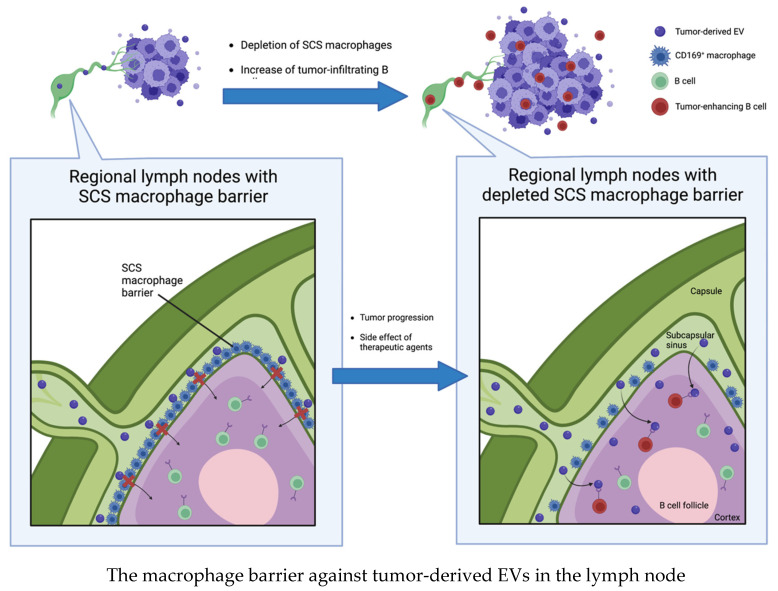
Subcapsular sinus CD169^+^ macrophages block tumor-derived EVs dissemination in tumor-draining lymph nodes. Once the barrier is disrupted by tumor progression or side effects of therapeutic agents, tumor-derived EVs enter the lymph node cortex and interact with B cells to induce tumor progression.

**Figure 4 ijms-24-01362-f004:**
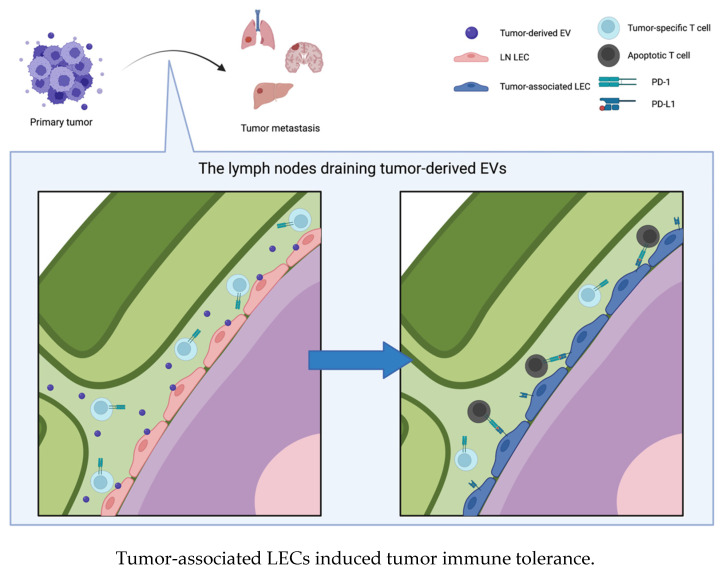
Tumor-derived EVs interact with lymph node resident lymphatic endothelial cells (LNLEC) and induce expression of PD-L1 on the surface of LNLEC. The tumor-associated LECs induce apoptosis of tumor-specific T cells by binding PD-1 on their surface, which results in tumor immune tolerance and tumor progression.

**Table 1 ijms-24-01362-t001:** EV-derived bioactive molecules associated with lymphangiogenesis or lymph node metastasis.

EV-Derived Bioactive Molecule	Type of Bioactive Molecule	Mechanism	Functional Effect	Cancer Type	Specimen	Isolation of EVs	Reference
miR-221-3p	miRNA	ERK/Akt pathway	Promoting lymphangiogenesis Promoting lymph node metastasis	CSCC	Serum supernatant of culture medium	ExoQuick^TM^ (SBI, America)	[96]
uc.189	ncRNA	Activating P38MAPK/VEGF-C pathway	Promoting lymphangiogenesis Promoting lymph node metastasis	ESCC	Supernatant of culture medium	ExoQuick^TM^ (SBI, America)	[97]
MIDKINE	protein	Regulating mTOR pathway	Promoting lymphangiogenesis	Melanoma	Supernatant of culture medium	Ultracentrifugation	[98]
ELNAT1	ncRNA	Activating hnRNPA1/UBC9/SOX18 pathway	Promoting lymphangiogenesis Promoting lymph node metastasis	Bladder cancer	Supernatant of culture medium	Ultracentrifugation	[99]
HANR	ncRNA	Activating miR-296/EAG1/VEGF pathway	Promoting lymphangiogenesis	HCC	Supernatant of culture medium	Ultracentrifugation	[100]
miR-320b	miRNA	Activating Akt pathway	Promoting lymphangiogenesis	ESCC	Supernatant of culture medium	Not listed	[101]
LNMAT-2	ncRNA	Upregulating PROX expression	Promoting lymphangiogenesis Promoting lymph node metastasis	Bladder cancer	Urine Supernatant of culture medium	Not listed	[102]
S100A8, S100A9	protein	Regulating CD83, CD86	Modulating DC maturation Establishing premetastatic niche in lymph nodes	Melanoma	Supernatant of culture medium	Total Exosome Isolation Reagent^TM^ (ThermoFisher, America)	[112,113]
miR-1468-5p	miRNA	Suppressing HMBOX-SOCS1 expression Activating JAK2/STAT3 pathway	Upregulating PD-L1 expression in LEC Promoting lymphangiogenesis	CSCC	Serum supernatant of culture medium	Ultracentrifugation	[130]
hsa_circ_0026611	circRNA	Regulating endocytosis pathway	not listed	ESCC	Serum	ExoQuick^TM^ (SBI, America)	[154]

miRNA, micro RNA; ncRNA, non-coding RNA; circRNA, circular RNA; DC, dendritic cell; PD-L1, programmed death ligand 1; LEC, lymphatic endothelial cell; CSCC, cervical squamous cell carcinoma; ESCC, esophageal squamous cell carcinoma; HCC, hepatocellular carcinoma.

## Data Availability

Not applicable.

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
