# Peer review of "Extracellular Vesicles Are Important Mediators That Regulate Tumor Lymph Node Metastasis via the Immune System"

_ijms, 2023, doi:10.3390/ijms24021362_

Round 1

Reviewer 1 Report

In this paper, Kiya et al. reviewed the relationship between EVs and tumor progression via the immune system as well as the clinical application of EVs as biomarkers and therapeutic agents. Moreover, the authors discussed the role of EVs in the development of the lymphatic system and tumor lymphatic metastasis. Overall, the authors provided valuable insights of EVs in cancer metastasis, yet there are few concerns listed below:

1. Please further discuss how EVs enhance the malignancy of tumor cells in section 2.1.

2. Figure 1 needs to be reedited. The core of Figure 1 is that tumor derived EVs enhance the malignancy of the cells, but the figure does not clearly show the detailed information to promote the malignant transformation of cells. In Figure 1, it is mentioned that tumor derived EVs influence variant cells by autocrine or parallel manager, but only tumor cells and fibroblasts are displayed, please show other related cells. In addition, please explain the meaning of cells with different colors.

3. In Figure 2, purple fibroblasts in Figure 2 should represent CAF, and TAM has no full name.

4. In Line 259, the barrier is weakened by tumor progression or the side effects of therapeutic agents. Authors should further explained the corresponding regulatory mechanism.

5. Figure 3, The left figure and the right figure should be marked with tumor barrier function or tumor promotion function respectively, and the conditions causing changes should be marked on the arrows in the middle. In addition, do the corresponding macrophages disappear in the tumor progression?

6. Lines 472-473, EVs and their cargoes could be stably preserved for over 90 days under

general storage conditions; Lines 494-495, uc.189 in EVs promotes the proliferation, migration, and tube formation of HLECs by activating the P38MAPK/VEGF-C pathway by binding to EPHA2 , please provide references.

7. Font size of Conclusions section is inconsistent with the others.

8. The manuscript should be further edited to minimize language problems. Following are some of the examples:

--Ly6Clow, CD4+, CD8+, pote4ntial;

--Lines 52-54, A phase II clinical trial investigated the clinical benefit of exosomes that were derived from IFN-γ-matured dendritic cells and loaded with MHC class I- and II-restricted cancer antigens as maintenance immunotherapy after induction chemotherapy in patients with inoperable non-small cell lung cancer without tumor progression;

--Lines 250-252: Furthermore, tumor-derived EVs induce the transformation of immune cells from cells that eliminate tumor cells to cells that promote tumor proliferation and metastatic potential.

--“Lines 252-254, tumor-derived EVs induce the polarization of macrophages toward the M2 phenotype. EVs derived from ovarian cancer, ESCC, glioblastoma, and other cancers induce the polarization of macrophages toward the M2 phenotype via various pathways.” Repetitively express the same point of view, and these two sentences should be integrated into one sentence.

Author Response

Reviewer 1

Comment: In this paper, Kiya et al. reviewed the relationship between EVs and tumor progression via the immune system as well as the clinical application of EVs as biomarkers and therapeutic agents. Moreover, the authors discussed the role of EVs in the development of the lymphatic system and tumor lymphatic metastasis. Overall, the authors provided valuable insights of EVs in cancer metastasis, yet there are few concerns listed below:

Response: Thank you for your clear summary and insightful comments on our manuscript. We feel the comments have helped us significantly improve this paper. We replied to your comments below respectively.

Query 1: Please further discuss how EVs enhance the malignancy of tumor cells in section 2.1. 

Response 1: Thank you for your valuable comments. In this section, we mainly discussed tumor-derived EVs enhance the tumor malignancy but we did not mention EVs derived from tumor microenvironmental cells enough. So, we added further description that CAF-derived EVs can promote tumor malignancy (line 150-155 in revised manuscript).

Then, we described that tumor-derived EVs enhance the malignancy of adjacent cells like low malignant potential tumor cells or tumor microenvironmental cells by autocrine/paracrine manner. In the end of this section, we have added the suggestion that tumor-derived EVs can influence distant organ or immune system by endocrine manner and we discussed that in next section. We could have made the connection to next section smooth by discussing other mechanism how EVs enhance the malignancy of tumor cells. We added the sentences as below (line 162-167 in revised manuscript). "Although we discussed that EVs enhance the tumor malignancy by interacting directly with tumor cells or tumor microenvironmental cells in this section, EVs can also promote tumor progression indirectly by influencing distant organs or immune system. Thus, EVs can mediate not only local effects around tumor as autocrine/paracrine manner but also systemic effects via blood circulation as endocrine manner. We have reviewed how EVs promote the tumor metastasis in distant organs next section."

Query 2: Figure 1 needs to be reedited. The core of Figure 1 is that tumor derived EVs enhance the malignancy of the cells, but the figure does not clearly show the detailed information to promote the malignant transformation of cells. In Figure 1, it is mentioned that tumor derived EVs influence variant cells by autocrine or parallel manager, but only tumor cells and fibroblasts are displayed, please show other related cells. In addition, please explain the meaning of cells with different colors.

Response 2: Thank you for your accurate suggestion. In Figure 2, it means that purple colored cells are high malignant cells, and the EVs received cells are altered their character worse as changing fibroblast cells to CAF. We reedited Figure 1 adding other related cells in tumor microenvironment (immune cells and endothelial cells) and appended the figure legend about different colored cells.

Query 3: In Figure 2, purple fibroblasts in Figure 2 should represent CAF, and TAM has no full name.

Response 3: Thank you for your accurate indication, and we are sorry for not explain about TAM. We renamed purple fibroblasts CAF and revised the name of TAM as Tumor-associated macrophage in Figure 2.

Query 4: In Line 259, the barrier is weakened by tumor progression or the side effects of therapeutic agents. Authors should further explained the corresponding regulatory mechanism.

Response 4: Thank you for your suggestion. In the reference article, the authors showed that cancer cells or chemotherapeutic agents (paclitaxel, carboplatin and CSF1-R inhibitor) decreased CD169+ SCS macrophage density but they did not provide the corresponding regulatory mechanism. This is because, we added the description that the detail mechanism of weakening the macrophage barrier was not evaluated in the reference article (line 280-281 in revised manuscript).

Query 5: Figure 3, The left figure and the right figure should be marked with tumor barrier function or tumor promotion function respectively, and the conditions causing changes should be marked on the arrows in the middle. In addition, do the corresponding macrophages disappear in the tumor progression?

Response 5: Thank you for your valuable comments. We added more detailed description of the barrier function in left of Figure 3. As you indicated, CD169+ SCS macrophages does not disappear, so we added few macrophages and described the disruption of the macrophage barrier in right of Figure 3. Further, we described the development of tumor-enhancing B cell and tumor progression by these B cells.

Query 6: “Lines 472-473, EVs and their cargoes could be stably preserved for over 90 days under general storage conditions”; “Lines 494-495, uc.189 in EVs promotes the proliferation, migration, and tube formation of HLECs by activating the P38MAPK/VEGF-C pathway by binding to EPHA2” , please provide references.

Response 6: We are sorry for not providing reference articles of these description and thank you for your accurate indicates. We added references of these sentences (line 496-497 and line 518-519 in revised manuscript).

Query 7: Font size of Conclusions section is inconsistent with the others.

Response 7: We are sorry for not create our manuscript according to manuscript template and thank you for your accurate indication. We revised the font size of Conclusions section.

Query 8: The manuscript should be further edited to minimize language problems. Following are some of the examples:

Response 8: Thank you for your accurate indication, we revised these errors respectively.

--Ly6Clow, CD4+, CD8+, pote4ntial;

We revised these descriptions (line 206, line 397, and line 482 in revised manuscript).

--Lines 52-54, A phase II clinical trial investigated the clinical benefit of exosomes that were derived from IFN-γ-matured dendritic cells and loaded with MHC class I- and II-restricted cancer antigens as maintenance immunotherapy after induction chemotherapy in patients with inoperable non-small cell lung cancer without tumor progression;

We corrected a grammatical error of this sentence (line 56 in revised manuscript).

--Lines 250-252: Furthermore, tumor-derived EVs induce the transformation of immune cells from cells that eliminate tumor cells to cells that promote tumor proliferation and metastatic potential.

We revised this sentence concisely as follows, "Furthermore, tumor-derived EVs induce the transformation of immune cells from tumor eliminating phenotype to tumor promoting phenotype." (line 264-265 in revised manuscript).

--“Lines 252-254, tumor-derived EVs induce the polarization of macrophages toward the M2 phenotype. EVs derived from ovarian cancer, ESCC, glioblastoma, and other cancers induce the polarization of macrophages toward the M2 phenotype via various pathways.” Repetitively express the same point of view, and these two sentences should be integrated into one sentence.

These sentences are mere repetition as you indicated. We deleted former sentence (line 272-273 in revised manuscript).

Reviewer 2 Report

This is a very concise review of the current scientific work on EV and LN tumor metastasis. I have no further comments.

Author Response

Reviewer 2

Comment: This is a very concise review of the current scientific work on EV and LN tumor metastasis. I have no further comments.

Response: We wish to express our strong appreciation to you for the kindly comment on our paper.

Reviewer 3 Report

The manuscript entitles "Extracellular vesicles are important mediators that regulate tumor lymph node metastasis via the immune system" by Yoshitaka Kika et al. is a very interesting and comprehensive review of the different implications of EVs in the tumorigenic and metastatic processes, with particular attention to the interplay of EVs and the immune system. The reviewer has only a couple of minor comments to this complete review:

-Authors should include tables to resume the major findings discussed in the manuscript

-The conclusions should be implemented with author's point of view and comment on future perspective

Author Response

Reviewer 3

Comment: The manuscript entitles "Extracellular vesicles are important mediators that regulate tumor lymph node metastasis via the immune system" by Yoshitaka Kika et al. is a very interesting and comprehensive review of the different implications of EVs in the tumorigenic and metastatic processes, with particular attention to the interplay of EVs and the immune system. The reviewer has only a couple of minor comments to this complete review:

Response: Thank you for the clear summary and accurate comments. We reply to your comments as follows, respectively.

Query 1: -Authors should include tables to resume the major findings discussed in the manuscript

Response 1: Thank you for your accurate suggestion, we added table resume EVs-derived bioactive molecules associated with lymphangiogenesis or lymph node metastasis (p.13).

Query 2: -The conclusions should be implemented with author's point of view and comment on future perspective

Response 2: We agree with your comments. We added our point of view and comment on future perspective as follows. "Thus, we expect that EVs can be novel therapeutic agents including immunological effects for cancer, which are completely different from existing therapeutic agents. We also expect EVs the preventive effect for tumor metastasis of highly metastatic tumor because of their closely association of establishing premetastatic niche." (line 609-612 in revised manuscript) And following sentences (line 614-617 in revised manuscript) are also our opinion but we are sorry for unclear sentences.

Reviewer 4 Report

The authors accurately discussed the role of extracellular vesicles in tumor progression through the immune system.

The review is very well written and dissected. The use of references is adequate to the context. The english is correct. 

I reccomend to accept the review in the present version.

Author Response

Reviewer 4

Comment: The authors accurately discussed the role of extracellular vesicles in tumor progression through the immune system.

The review is very well written and dissected. The use of references is adequate to the context. The english is correct. 

I reccomend to accept the review in the present version.

Response: We wish to express our appreciation to you for the kindly comments on our paper.

Round 2

Reviewer 1 Report

The authors have responsed to the comments point by point and revised the manuscript. This paper is worth to be published.